# OpenReview forum: "Curriculum-Guided Layer Scaling for Language Model Pretraining"
_ICML.cc/2026/Conference — ICML 2026 regular_

### Official Review · Reviewer_b8fb · 2026-03-10

**Soundness:** 3
**Presentation:** 3
**Significance:** 2
**Originality:** 2
**Overall Recommendation:** 5
**Confidence:** 4

**Summary:**

The paper investigates combining curriculum learning with layer scaling for language models.
Language model pre-training is performed in stages, with each stage being trained on increasingly more complex data and incorporating more layers.
During the training of the new parameters in each stage, the old ones are first frozen for a period, in order to optimize the randomly initialised parameters to a reasonable state first.

**Compliance With Llm Reviewing Policy:**

Affirmed.

**Key Questions For Authors:**

In the "Layer-scaling only" baseline, are the old layers also initially frozen, same as CGLS?

**Limitations:**

Yes

**Strengths And Weaknesses:**

The method proposed is logical and intuitive.
There are a good number of experiments, different scales, different ablated baselines and different downstream tasks.
The results at the smaller scale are somewhat mixed, but the results at the largest scale show a more convincing improvement.

The proposed method is somewhat incremental, combining two established techniques together. Although the experiments are thorough and it is interesting to see that this combination seems to work well together, outperforming each method individually.

It is said that the FLOP budget is fixed for all experiments. It is unclear how this FLOP budget was chosen. For example, it may have been chosen based on the performance of the proposed method, which would be unfair.

The baselines not using layer scaling will presumably use more FLOPs, so that seems to mean that these approaches will be shown less data, in order to fit the same FLOP budget?
In that case, the performance differences could be the result of the baselines being trained on less data.
That would mean the contribution of this approach is mainly in the efficiency of the training process, however, at the moment the paper mainly emphasises performance improvements. This seems a bit misleading.

The Domain-Adaptive Code Pretraining experiment is not useful in its current form and is misleading. It attributes the domain adaptation benefit to the proposed method. But training on domain-specific data in the last stage is a very well established approach for training domain-specific models.
The experiment needs to report Baseline (Curricularized) in order to have any basis to claim some benefit from the proposed method.

Instead, an experiment without freezing the old layers when introducing new ones would be interesting, to determine how important this is to the training process.

The equation at the bottom of page 3 is confusing. The meaning of the circ operator is not defined and most of the parameter groups are not explained either.

---

> ### Author Rebuttal · Authors · 2026-03-31
>
> We thank the reviewer for the positive evaluation and are glad that you found the method “intuitive” and the experimental setup thorough. Below we address your questions and concerns.
>
> ---
>
> > “The FLOP budget... may have been chosen based on the performance of the proposed method.”
>
> Thank you for raising this concern. We would like to clarify that the FLOP budget is not tuned based on the performance of our method. Instead, it is fully determined by the choice of model architecture (e.g., GPT-2-Small, LLaMA-3.2-1B) and the number of training tokens (e.g., 700M, 2.5B, 20B), which are fixed *a priori*. All methods are trained under this same compute budget with identical token counts and model configurations, ensuring a fair comparison.
>
> ---
>
> > “Baselines... will be shown less data in order to fit the same FLOP budget… performance differences could be [due to data exposure].”
>
> All methods are trained on the same number of tokens and dataset, and reach the same final model size. The FLOP budget is therefore identical, and the difference lies only in how compute is allocated: baselines train the full model for all tokens, whereas CGLS allocates some compute to building up the model from a smaller depth (therefore actually seeing less tokens at the final model depth). We agree that this highlights an efficiency aspect of the method, and we will clarify in the paper that CGLS improves both compute efficiency and downstream performance under a fixed training budget.
>
> ---
>
> > “The domain-adaptive code experiment is not useful in its current form… needs to report [a curricularized] baseline.”
>
> We appreciate this suggestion and now include a curricularized baseline (DCLM to StarCoder), which underperforms the randomized baseline. We attribute this to the dynamics of domain-adaptive pretraining. In this new baseline, the model is exposed to code only in the second half of training and experiences a sudden domain shift, which limits the total effective optimization on the target distribution and requires adapting representations that are already biased toward natural language. In contrast, the randomized baseline sees code throughout, allowing it to gradually learn a joint distribution. CGLS improves over both baselines by adding new layers when transitioning to code, reducing interference with earlier representations and enabling gradual adaptation.
>
> | Benchmark  | Metric    | Baseline (Rand) | Baseline (Curr) | CGLS      |
> |------------|-----------|------------------|------------------|-----------|
> | HumanEval  | pass @ 8  | 3.56%           | 2.24%           | **6.67%** |
> |            | pass @ 16 | 4.26%           | 2.49%           | **7.63%** |
> |            | pass @ 32 | 5.27%           | 2.74%           | **8.78%** |
> |            | pass @ 64 | 6.71%           | 3.05%           | **9.76%** |
>
> ---
>
> > “An experiment without freezing old layers would be interesting.”
>
> We thank the reviewer for this suggestion and conducted an additional experiment removing the freezing/initialization phase. The results are shown below.
>
> | Model            |  PIQA | ARC-E | ARC-C | HellaSwag | Lambada |  Avg  |
> |------------------|:-----:|:-----:|:-----:|:---------:|:-------:|:-----:|
> | Baseline         | 59.09 | 36.36 | 24.24 |   34.20   |  **32.99**  | 37.78 |
> | CGLS (Paper)     | 61.21 | 41.37 | 25.43 |   **34.65**   |  32.74  | **39.09** |
> | CGLS (No Freeze) | **62.02** | **42.21** | 24.57 |   31.28   |  27.21  | 37.46 |
>
> We observe that removing freezing leads to mixed behavior: performance improves slightly on some benchmarks (e.g., PIQA, ARC-E), but degrades substantially on others (e.g., HellaSwag, Lambada), leading to worse average accuracy. We interpret this as a stability–plasticity tradeoff: removing freezing improves rapid adaptation but increases interference with earlier representations. Supporting this, the perplexity of the final models on the first stage’s data mixture is 26.96 (baseline), 24.50 (CGLS), and 30.84 (No Freeze), indicating worse retention of earlier learned representations without freezing.
>
> ---
>
> > The equation at the bottom of page 3 is confusing. The meaning of the circ operator is not defined and most of the parameter groups are not explained either.
>
> Thank you for pointing this out! We agree that the notation in this equation is underspecified and can be clarified. The operator $\circ$ denotes concatenation of parameter groups corresponding to different components of the model. We will revise the text to include these details and improve clarity.
>
> ---
>
> > In the "Layer-scaling only" baseline, are the old layers also initially frozen, same as CGLS?
>
> Yes, every component of the layer-scaling only baseline is identical to CGLS, aside from the removal of the data curriculum component.
>
> --
>
> We thank the reviewer again for the constructive feedback and suggestions. We will incorporate these clarifications and additional comparisons to further strengthen the presentation and isolate the contributions of CGLS.

---

> > ### Author Rebuttal · Reviewer_b8fb · 2026-04-03
> >
> > Thank you for your response.
> >
> > > All methods are trained under this same compute budget with identical token counts and model configurations, ensuring a fair comparison.
> >
> > That seems contradictory. FLOP budget is not the same as token count. If the method changes the size of the model during training, then how can it possibly have both the same FLOP count and the same token count as a baseline model with the same final size for the full training duration?
> >
> > > An experiment with curricularized baseline.
> >
> > Something seems to have gone wrong there, as it shows not only a non-improvement, but a major drop over the random baseline. This goes against a lot of previous work on curriculum learning. Aspects like learning rates and the length of different stages can be important, none of which were specified.
> >
> > I will keep my current score.

---

> > > ### Author Response · Authors · 2026-04-04
> > >
> > > Thank you for the follow-up.
> > >
> > > Regarding the compute budget: to clarify our setup, we fix the total number of training tokens sampled from the dataset across all methods, and define the compute budget / total FLOPs accordingly. Indeed, in earlier stages, smaller models will process more tokens for the same number of FLOPs, and therefore the CGLS experiments will have some repetition of the data compared to the baseline. However, they do not see any _additional_ data compared to the baselines.
> > >
> > > As for the curricularized domain-adaptation experiment, we agree that performance can be sensitive to factors such as stage length and optimization hyperparameters. In our setup, we used equal-length stages (~2B tokens of DCLM followed by ~2B tokens of StarCoder data, using the same datasets and splits as the randomized baseline), and reduced the learning rate by 2x in the second code stage (from 2e-4 to 1e-4). Alternate variants such as maintaining the learning rate throughout yielded similar results.

---

### Official Review · Reviewer_qP6C · 2026-03-11

**Soundness:** 3
**Presentation:** 3
**Significance:** 3
**Originality:** 2
**Overall Recommendation:** 4
**Confidence:** 3

**Summary:**

This paper proposes Curriculum-Guided Layer Scaling (CGLS), a pretraining framework that coordinates model depth expansion with increasing data difficulty. CGLS utilizes staged training, where newly added layers are first trained in isolation (with earlier layers frozen), before the entire model is fine-tuned on a more complex data distribution. The authors conduct experiments at multiple scales (GPT-2-Small, LLaMA-3.2-1B) and show that CGLS outperforms several baselines.

**Compliance With Llm Reviewing Policy:**

Affirmed.

**Final Justification:**

Most of my concerns have been addressed (train with more tokens)

**Key Questions For Authors:**

Please see the weaknesses.

**Limitations:**

yes

**Strengths And Weaknesses:**

### Strengths
1. The combination of curriculum learning and progressive layer scaling is novel and reasonable.
2. Write clearly and easily understandable.

### Weaknesses
1. The experiments using 700M tokens for the GPT-2-Small scale and 2.5B tokens for the Llama-3.2-1B scale cannot guarantee sufficient pre-training. As for the 20B-token setting, where the experiments are insufficient, all main experiments should be conducted under this setting (20× more tokens than parameters).
2. The results of MIDAS in Table 1 are even worse than the Baseline (Randomized), which is abnormal. Does this mean that MIDAS is actually ineffective?
3. The core insight of the paper is the integration of curriculum learning and progressive layer scaling into a unified pretraining framework. However, it does not conduct in-depth experimental analysis of progressive layer scaling (such as the impact of the initial number of layers on the final results), and focuses more on curriculum learning in terms of data.

---

> ### Author Rebuttal · Authors · 2026-03-31
>
> We thank the reviewer for their helpful feedback and are glad that you found the core idea “novel” and the paper “easily understandable.” Below we address your concerns.
>
> ---
>
> > “The experiments… cannot guarantee sufficient pre-training… all main experiments should be conducted under this setting (20× more tokens than parameters)."
>
> We agree that larger-scale training (e.g., 20B tokens for a 1B model) is important for evaluating absolute model quality.  To address this for GPT-2-Small, we additionally ran a larger-budget experiment (2.5B tokens / 20x ratio), where we observe consistent improvements from CGLS over the baselines in the table below. Importantly, in our 1B-parameter, 20B-token experiments, the gains from CGLS persist and are in fact larger, supporting the scalability of the approach. We therefore view smaller-scale experiments as controlled settings for isolating effects, and larger-budget experiments as validation in more realistic regimes.
>
> | Model              |  PIQA | ARC-E | ARC-C | HellaSwag | Lambada |  Avg  |
> |--------------------|:-----:|:-----:|:-----:|:---------:|:-------:|:-----:|
> | Baseline (Rand)    | *60.94* | 38.85 | *23.38* |   26.89   |  25.58  | 35.13 |
> | Baseline (Curr)    | 60.61 | *40.03* | 22.53 |   27.17   |  26.12  | 35.29 |
> | Layer-Scaling Only | 60.23 | 38.43 | 21.93 |   27.05   |  27.05  | 34.94 |
> | MIDAS              | 60.77 | 37.58 | **24.40** |   *26.77*   |  **28.29**  | *35.56* |
> | CGLS               | **61.64** | **43.01** | *23.12* |   **27.21**   |  *28.08*  | **36.61** |
>
> ---
>
> > “MIDAS results... are even worse than the baseline… does that mean that MIDAS is actually ineffective?”
>
> Thank you for raising this point. The original paper (Saunshi et al., 2024) shows that MIDAS exhibits task-dependent behavior, underperforming on memorization-focused tasks while improving certain reasoning benchmarks. A similar pattern appears in our results: MIDAS performs strongly on some tasks (e.g., ARC-C, Lambada) but is not consistently better across all benchmarks or in the overall average. This highlights a limitation of stacking-only approaches: while they can improve specific capabilities, their benefits are not uniform. Our goal is to address this inconsistency. By introducing a curriculum over data complexity, CGLS achieves more consistent improvements across benchmarks, yielding the strongest overall average while remaining competitive on individual tasks. This suggests that synchronization between model growth and data complexity, and not stacking alone, is key to broader gains.
>
> ---
>
> > “The paper... does not conduct in-depth experiment analysis of progressive layer scaling…”
>
> We agree that understanding the role of layer scaling is important. Due to space constraints, we placed detailed analyses in Appendix A.3, including ablations over initial depth and stage-wise compute allocation. These experiments show clear trends, for example that starting at approximately half the final depth yields the best performance. To further isolate the effect of layer scaling, we include a layer-scaling-only baseline in Table 1 of the paper, which uses the same progressive stacking procedure without a curriculum. This baseline underperforms CGLS, demonstrating that layer scaling alone is insufficient and that synchronization with data complexity is critical. We will make these points more visible in the main text.
>
> ---
>
> We appreciate the reviewer’s feedback and help in improving our paper. We will revise the paper to (1) clarify the role of scale and experimental design, and (2) better highlight the contributions of progressive layer scaling and synchronization.

---

> > ### Author Rebuttal · Reviewer_qP6C · 2026-04-03
> >
> > Most of my concerns have been addressed

---

> > > ### Author Response · Authors · 2026-04-06
> > >
> > > We appreciate your thoughtful review and consideration! We’re glad that our responses helped resolve your concerns, and hope this will be reflected in your final evaluation.

---

### Official Review · Reviewer_mrPB · 2026-03-15

**Soundness:** 3
**Presentation:** 3
**Significance:** 3
**Originality:** 2
**Overall Recommendation:** 4
**Confidence:** 4

**Summary:**

This paper studies whether progressive model growth can be made more effective for language model pretraining by coordinating it with a curriculum over data complexity. The proposed method, Curriculum-Guided Layer Scaling (CGLS), expands model depth in stages and pairs each stage with a progressively harder data mixture. When new layers are added, they are first trained in isolation and then the full model is jointly tuned. Across compute-matched experiments at GPT-2-Small and Llama-3.2-1B scales, plus a 20B-token extension and a web-to-code domain-shift setting, the paper reports that CGLS consistently outperforms randomized training, curriculum-only training, and stacking-only baselines on several downstream evaluations.

**Compliance With Llm Reviewing Policy:**

Affirmed.

**Key Questions For Authors:**

- Can the authors provide more intuition or evidence for why CGLS hurts perplexity at the 2.5B-token scale while improving downstream performance?

**Limitations:**

yes

**Strengths And Weaknesses:**

Strengths:

- The core hypothesis is clear and well motivated. The paper does not simply claim that progressive stacking helps; it argues that stacking becomes useful when the model's increasing capacity is aligned with a structured increase in data complexity.
- The authors include compute-matched randomized, curriculum-only, layer-scaling-only, and MIDAS baselines, which makes it easier to isolate what CGLS is actually contributing.
- In addition to the main GPT-2-Small and 1B-scale runs, the paper includes a 20B-token extension, a domain-shift experiment from web text to code, seed-replication results, hyperparameter ablations, and a robustness check for the difficulty classifier.
- Presentation is generally solid. The paper is easy to follow, the staged training procedure is clearly described, and the appendix contains enough implementation detail to understand the schedules and major hyperparameters.

Weaknesses:
- All experiments stop at roughly the 1B-parameter regime, so it remains unclear whether the same gains persist at the larger scales where modern pretraining decisions matter most in practice.
- Some of the gains are modest in the smaller experiments. At GPT-2-Small scale, CGLS improves the average score only slightly over the curricularized baseline (36.46 vs 36.12), which suggests that the full method is not yet clearly stronger than simpler curriculum training in low-resource settings.
- Some results are mixed across tasks. For example, at 20B tokens CGLS underperforms the curricularized baseline on OpenRewriteEval, and at the 2.5B-token scale it is only tied or slightly worse on some tasks such as HellaSwag and Lambada. The method therefore seems more beneficial for certain reasoning- and knowledge-heavy tasks than as a universal pretraining improvement.

---

> ### Author Rebuttal · Authors · 2026-03-31
>
> We thank the reviewer for their helpful and balanced evaluation. We are glad that you found the core hypothesis “clear and well motivated”, and that you appreciated the breadth of our experimental design. Below we address your concerns in detail.
>
> ---
>
> | “All experiments stop at roughly the 1B-parameter regime…”
>
> We agree that evaluation at larger parameter scales (e.g., 7B+) would be valuable. However, our goal in this work is to study training dynamics under controlled, compute-matched settings, rather than to maximize absolute model performance. The 1B-parameter regime enables systematic ablations over curriculum design, model growth schedules, and stage allocations—analyses that are infeasible at larger scales under academic compute constraints.
>
> Importantly, we provide evidence of scalability along two axes. First, we include a Chinchilla-optimal experiment (1B, 20B tokens), where gains from CGLS persist and are in fact larger than at smaller budgets. Second, we evaluate a domain-shift setting (general text to code), where CGLS also yields consistent improvements. Together, these results suggest that the benefits of synchronizing model growth with data complexity are not limited to small-scale regimes. We will clarify this motivation more explicitly in the revision.
>
>
> ---
>
> | “Some of the gains are modest in the smaller experiments…”
>
> We view the GPT-2-Small experiments as a controlled proof-of-concept setting. At this scale, all methods are strongly capacity-limited, and improvements are necessarily modest. Importantly, the gap between CGLS and curriculum-only baselines widens at larger scales (1B and 20B tokens), suggesting that the benefits of synchronization grow with model capacity and training budget. We will make this scaling trend more explicit in the paper.
>
> ---
>
> | “Some results are mixed across tasks…”
>
> We agree that CGLS does not uniformly improve all tasks. In our experiments, gains are most pronounced on reasoning- and knowledge-intensive benchmarks (e.g., ARC, PIQA, GPQA), while performance remains comparable on tasks such as HellaSwag, Lambada, and OpenRewriteEval, depending on the scale. We view this as evidence that CGLS primarily improves representations relevant for structured reasoning and transfer, rather than uniformly optimizing next-token prediction across all domains. Notably, CGLS achieves the strongest overall average performance, indicating more balanced improvements across benchmarks even when individual tasks vary.
>
> ---
>
> | “Why [does] CGLS hurt perplexity at 2.5B tokens while improving downstream performance?”
>
> We hypothesize that this reflects a known disconnect between next-token likelihood and downstream task performance. Prior work has shown that architectural inductive biases, in particularly depth, can disproportionately affect transfer performance even when perplexity is unchanged or worse. Since CGLS explicitly modifies model depth during training, it may bias the model toward representations that transfer better to structured reasoning tasks, even if they are not optimal under likelihood-based objectives. Importantly, this effect diminishes at larger scale: in our 20B-token experiment, CGLS improves both perplexity and downstream performance, suggesting that these objectives better align with sufficient training.
>
> ---
>
> Again, we appreciate the reviewer’s feedback. We will revise the paper with these clarifications, and hope they further strengthen the case for CGLS as a principled and scalable approach to curriculum-guided pretraining.

---

> > ### Author Rebuttal · Reviewer_mrPB · 2026-04-02
> >
> > I appreciate the authors' rebuttal and maintain my current positive score.

---

### Official Review · Reviewer_D3H2 · 2026-03-17

**Soundness:** 3
**Presentation:** 2
**Significance:** 4
**Originality:** 3
**Overall Recommendation:** 4
**Confidence:** 4

**Summary:**

This paper proposes CGLS, the core idea of which is to synchronize the gradual stacking of the model with the data curriculum, hoping to make progressive layer scaling a more effective pretraining strategy. The paper points out that individual stacking is not enough and must be accompanied by a gradually more difficult data distribution. From the results, the paper indeed gives some positive signals: under the compute-matched setting, CGLS outperforms several baselines such as randomized, curricularized, layer-scaling-only, and MIDAS on multiple downstream tasks; the most eye-catching is the domain-adaptive code pretraining experiment.

**Compliance With Llm Reviewing Policy:**

Affirmed.

**Key Questions For Authors:**

See above.

**Limitations:**

Yes

**Strengths And Weaknesses:**

Strengths

1. The research question is valuable. This paper is not just adding another training trick, but is asking a more critical question: why does progressive stacking often fall short of full-capacity training, and whether it can be improved by synchronizing with a data curriculum.

2. The method is intuitively clear, and the overall story holds together: the narrative of CGLS is coherent, with model capacity gradually increasing and data complexity also gradually rising; new layers are first trained separately and then integrated into the full model for joint training. This design is at least conceptually self-consistent and is indeed more principled than simply adding layers.

3. The experiment of transitioning from DCLM to StarCoder2 for code pretraining is quite impressive. Compared to random mixing, CGLS changed it to a progressive curriculum from general text to code, resulting in improvements in all pass@k on HumanEval, with pass@8 nearly doubling.


Weaknesses

1. The paper currently seems more like validating a set of effective recipes rather than validating a general principle; this is my biggest reservation. The benefits of CGLS may come from many intertwined factors: the way of augmentation, freezing and thawing, budget allocation, initial layer number, data bucketing, and the training sequence itself. What the authors have proven now is more like this whole set of recipes is effective, but they have not yet strongly proven that the principle itself that truly plays a key role is 'the model growth should be synchronized with the growth of data complexity.

2. The paper does not truly show: whether the old distribution capacity is more stable before and after adding layers, whether forgetting is reduced, and whether early learned representations are really better preserved.

3. GPT-2 small experiments are more like a proof-of-concept, while the 1B scale is much better than toy settings, but it is still not enough to fully support the author's stronger framing, such as "new paradigm" or more general compute-efficient pretraining conclusions. Especially, the paper actually shows a less clean phenomenon under the 1B/2.5B-token setting: the performance on downstream tasks improves, but the perplexity is not necessarily better.

---

> ### Author Rebuttal · Authors · 2026-03-31
>
> We thank the reviewer for their thoughtful and constructive feedback. We are glad that you found the research question "valuable", the method “intuitively clear,” and the domain-adaptive code experiment "quite impressive". Below we address your concerns in detail and provide additional analyses.
>
> ---
>
> > “The paper currently seems more like validating a set of effective recipes rather than validating a general principle… [it is] not yet strongly proven that [synchronization is the key factor.]”
>
> We agree that disentangling the contribution of synchronization is important. Our design already isolates this effect: (1) curriculum-only and layer-scaling-only baselines keep all other factors fixed, and (2) CGLS differs only in synchronizing the two. The consistent gap between these baselines and CGLS suggests that neither component alone explains the gains.
>
> To further test this, we ran a desynchronization experiment where we reverse the curriculum order while keeping all other factors fixed, with results shown in the table below. If the gains were due to generic aspects of the training recipe (e.g., freezing/thawing, staged optimization), we would expect performance to remain comparable. Instead, performance drops substantially across all benchmarks, below both the baseline and CGLS. We interpret this as evidence that the benefits of CGLS arise specifically from **synchronizing model growth with increasing data complexity**. When this alignment is broken, the same training procedure becomes ineffective.
>
> | Model         | PIQA      | ARC-E     | ARC-C     | HellaSwag | Lambada   | Avg       |
> |---------------|-----------|-----------|-----------|-----------|-----------|-----------|
> | Baseline      | 59.09     | 36.36     | 24.24     | 34.20     | **32.99** | 37.37     |
> | CGLS (Paper)  | **61.21** | **41.37** | **25.43** | **34.65** | 32.74     | **39.08** |
> | CGLS (Desync) | 56.64     | 32.41     | 23.46     | 27.97     | 22.12     | 32.52     |
>
> ---
>
> > “The paper does not truly show... whether early learned representations are really better preserved.”
>
> We thank the reviewer for raising this point. To assess whether early representations are better preserved under CGLS, we evaluate the final model on data from both early and late curriculum stages. As shown below, CGLS consistently achieves lower perplexity than the baseline on both the earliest stage mixture and the subset of the data with the lowest complexity classification, indicating stronger retention of previously learned distributions. We interpret this as evidence that CGLS helps stabilize learned representations, allowing earlier-stage knowledge to be preserved more effectively as capacity increases.
>
> | Eval Subset       | Baseline PPL | CGLS PPL    | $\Delta$ (Baseline - CGLS) | Relative Gain |
> |-------------------|--------------|-------------|-----------------------------|---------------|
> | Stage 1 Mixture   | 26.9636      | **24.4975** | 2.4661                      | 9.15%    |
> | Simplest Examples | 27.9900      | **25.6431** | 2.3468                      | 8.38%   |
> | Stage 4 Mixture   | 28.6047      | **25.7281** | 2.8766                      | 10.06%  |
> | Hardest Examples  | 28.8622      | **25.8518** | 3.0103                      | 10.43%  |
>
> ---
>
> > “The 1B scale... is still not enough to support [stronger claims] such as a new paradigm… the 1B/2.5B-token setting shows a less clean phenomenon... and perplexity is not necessarily better.”
>
> We agree that larger experiments are important for evaluating absolute model quality. However, our goal is not to maximize performance at scale, but to study training dynamics under controlled, compute-matched settings. The 1B-parameter regime allows us to isolate the interaction between curriculum and model growth while enabling extensive ablations that would be infeasible at larger scales under academic compute constraints.
>
> Importantly, we also evaluate CGLS at a 20B-token Chinchilla-optimal setting, where gains are in fact larger, supporting the scalability of the approach. Regarding the observation that perplexity does not always improve, this aligns with prior work showing that perplexity is not always predictive of downstream performance, particularly when architectural inductive biases like model depth differ as in CGLS [1, 2]. We view this as an informative signal rather than a contradiction.
>
> [1] Tay et al. Scale Efficiently: Insights from Pre-training and Fine-tuning Transformers. ICLR 2022.
>
> [2] Tay et al. Scaling Laws vs Model Architectures: How does Inductive Bias Influence Scaling? EMNLP Findings, 2023.
>
> ---
>
> Overall, we appreciate the reviewer’s insightful comments. We have added (1) a desynchronization experiment to better isolate the role of synchronization, and (2) retention analyses to evaluate representation stability. Together, we hope these strengthen the evidence that CGLS is not merely a collection of heuristics, but a structured interaction between model growth and data complexity.

---

> > ### Author Rebuttal · Reviewer_D3H2 · 2026-04-03
> >
> > Thank you for the detailed response.
> >
> > I still feel the claim is somewhat strong. In particular, reversing the curriculum order is a fairly extreme intervention, so the result may not cleanly isolate synchronization as the key factor. For example, one could keep the curriculum order unchanged and instead vary the training strategy.
> >
> > I also think the 1B-scale analysis and the added retention results have value. However, it remains difficult to judge how well these findings would generalize at larger scale.
> >
> > I understand that this is partly due to resource constraints, so I do believe the paper is valuable overall, but some of the broader concerns cannot be fully resolved in the current version.

---

### Decision · Program_Chairs · 2026-04-30

**Decision:**

Accept (regular)

**Comment:**

The reviewers expressed some concerns in their initial reviews, which were addressed during the rebuttal. While the experiments still seem somewhat limited in scale (see the comment by Reviewer D3H2), all reviewers agree that the paper presents a promising approach. I therefore vote to accept the paper.